# From deep learning to mechanistic understanding in neuroscience: the structure of retinal prediction

**Hidenori Tanaka**[1,2,†], **Aran Nayebi**[3], **Niru Maheswaranathan**[3,5], **Lane McIntosh**[3], **Stephen A. Baccus**[4], **and Surya Ganguli**[2,5,†]

[1]Physics & Informatics Laboratories, NTT Research, Inc., East Palo Alto, CA, USA
[2]Department of Applied Physics, Stanford University, Stanford, CA, USA
[3]Neurosciences PhD Program, Stanford University, Stanford, CA, USA
[4]Department of Neurobiology, Stanford University, Stanford, CA, USA
[5]Google Brain, Google, Inc., Mountain View, CA, USA
[†]{tanaka8,sganguli}@stanford.edu

## Abstract

Recently, deep feedforward neural networks have achieved considerable success in modeling biological sensory processing, in terms of reproducing the input-output map of sensory neurons. However, such models raise profound questions about the very nature of explanation in neuroscience. Are we simply replacing one complex system (a biological circuit) with another (a deep network), without understanding either? Moreover, beyond neural representations, are the deep network's *computational mechanisms* for generating neural responses the same as those in the brain? Without a systematic approach to extracting and understanding computational mechanisms from deep neural network models, it can be difficult both to assess the degree of utility of deep learning approaches in neuroscience, and to extract experimentally testable hypotheses from deep networks. We develop such a systematic approach by combining dimensionality reduction and modern attribution methods for determining the relative importance of interneurons for specific visual computations. We apply this approach to deep network models of the retina, revealing a conceptual understanding of how the retina acts as a predictive feature extractor that signals deviations from expectations for diverse spatiotemporal stimuli. For each stimulus, our extracted computational mechanisms are consistent with prior scientific literature, and in one case yields a new mechanistic hypothesis. Thus overall, this work not only yields insights into the computational mechanisms underlying the striking predictive capabilities of the retina, but also places the framework of deep networks as neuroscientific models on firmer theoretical foundations, by providing a new roadmap to go beyond comparing neural representations to extracting and understand computational mechanisms.

## 1 Introduction

Deep convolutional neural networks (CNNs) have emerged as state of the art models of a variety of visual brain regions in sensory neuroscience, including the retina [1, 2], primary visual cortex (V1), [3, 4, 5, 6], area V4 [3], and inferotemporal cortex (IT) [3, 4]. Their success has so far been primarily evaluated by their ability to explain reasonably large fractions of variance in biological neural responses across diverse visual stimuli. However, fraction of variance explained is not of course the same thing as scientific explanation, as we may simply be replacing one inscrutable black box (the brain), with another (a potentially overparameterized deep network).

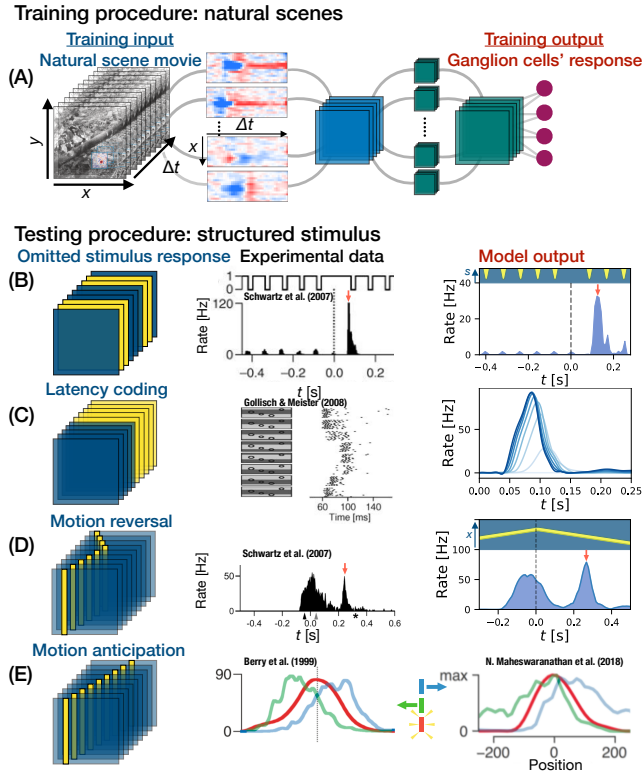

Training procedure: natural scenes

Training input
Natural scene movie

Training output
Ganglion cells' response

(A)

Testing procedure: structured stimulus

Omitted stimulus response   Experimental data

Model output

(B)

Latency coding

(C)

Motion reversal

(D)

Motion anticipation

(E)

Figure 1: **Deep learning models of the retina trained only on natural scenes reproduce an array of retinal phenomena with artificial stimuli (reproduced from ref. [2]).**

(A) Training procedure: We analyzed a three-layer convolutional neural network (CNN) model of the retina which takes as input a spatiotemporal natural scene movie and outputs a nonnegative firing rate, corresponding to a retinal ganglion cell response. The first layer consists of eight spatiotemporal convolutional filters (i.e., cell types) with the size of $(15 \times 15 \times 40)$, the second layer of eight convolutional filters $(8 \times 11 \times 11)$, and the fully connected layer predicting the ganglion cells' response. As previously reported in [2], the deep learning model reproduces (B) an omitted stimulus response, (C) latency coding, (D) the motion reversal response, and (E) motion anticipation.

Indeed, any successful scientific model of a biological circuit should succeed along three fundamental axes, each of which goes above and beyond the simple metric of mimicking the circuit's input-output map. First, the intermediate *computational mechanisms* used by the hidden layers to generate responses should ideally match the intermediate computations in the brain. Second, we should be able to extract *conceptual insight* into *how* the neural circuit generates nontrivial responses to interesting stimuli (for example responses to stimuli that cannot be generated by a linear receptive field). And third, such insights should suggest new experimentally testable hypotheses that can drive the next generation of neuroscience experiments.

However, it has been traditionally difficult to systematically extract computational mechanisms, and consequently conceptual insights, from deep CNN models due to their considerable complexity [7, 8]. Here we provide a method to do so based on the idea of model reduction, whose goal is to systematically extract a simple, reduced, minimal subnetwork that is most important in generating a complex CNN's response to any given stimulus. Such subnetworks then both summarize computational mechanisms and yield conceptual insights. We build on ideas from interpretable machine learning, notably methods of input attribution that can decompose a neural response into a sum of contributions either from individual pixels [9] or hidden neurons [10]. To achieve considerable model reduction for responses to spatiotemporal stimuli, we augment and combine such input attribution methods with dimensionality reduction, which, for carefully designed artificial stimuli employed in neurophysiology experiments, often involves simple spatiotemporal averages over stimulus space.

We demonstrate the power of our systematic model reduction procedure to attain mechanistic insights into deep CNNs by applying them to state of the art deep CNN models of the retina [1, 2]. The retina constitutes an ideal first application of our methods because the considerable knowledge (see e.g. [11]) about retinal mechanisms for transducing spatiotemporal light patterns into neural responses enables us to assess whether deep CNNs successfully learn the same computational structure. In particular, we obtain deep CNN models from [2] which were trained specifically to mimic the input-output transformation from natural movies to retinal ganglion cell outputs measured in the salamander retina. The model architecture involved a three-layer CNN model of the retina with ReLU nonlinearities (Fig. 1A). This network was previously shown [1, 2] to: (i) yield state of the art models of the retina's response to natural scenes that are almost as accurate as possible given intrinsic retinal stochasticity; (ii) possess internal subunits with similar response properties to those of retinal interneurons, such as

bipolar and amacrine cell types; (iii) generalize from natural movies, to a wide range of *eight* different classes of artificially structured stimuli used over decades of neurophysiology experiments to probe retinal response properties. This latter generalization capacity from natural movies to artificially structured stimuli (that were never present in the training data) is intriguing given the vastly different spatiotemporal statistics of the artificial stimuli versus natural stimuli, suggesting the artificial stimuli were indeed well chosen to engage the same retinal mechanisms engaged under natural vision [2].

Here, we focus on understanding the computational mechanisms underlying the deep CNN's ability to reproduce the neural responses to four classes of artificial stimuli (Fig. 1B-E), each of which, through painstaking experiments and theory, have revealed striking nonlinear retinal computations that advanced our scientific understanding of the retina. The first is the omitted stimulus response (OSR) [12, 13] (Fig. 1B), in which a periodic sequence of full field flashes entrains a retinal ganglion cell to respond periodically, but when a single flash is omitted, the ganglion cell produces an even larger response at the expected time of the response to the omitted flash. Moreover, the timing of this omitted stimulus response occurs at the expected time over a range of frequencies of the periodic flash train, suggesting the retina is somehow retaining a memory trace of the period of the train of flashes. The second is latency encoding [14], in which stronger stimuli yield earlier responses (Fig. 1C). The third is motion reversal [15], in which a bar suddenly reversing its motion near a ganglion cell receptive field generates a much larger response after the motion reversal (Fig. 1D). The fourth is motion anticipation [16], where the neural population responding to a moving bar is advanced in the direction of motion to compensate for propagation delays through the retina (Fig. 1E). These responses are striking because they imply the retina has implicitly built into it a predictive world model codifying simple principles like temporal periodicity, and the velocity based extrapolation of future position. The retina can then use these predictions to improve visual processing (e.g. in motion anticipation), or when these predictions are violated, the retina can generate a large response to signal that deviation (e.g. in the OSR and motion reversal).

While experimentally motivated prior theoretical models have been employed to explain the OSR [17, 18], latency encoding [14, 19], motion reversal [20, 21], and motion anticipation [16], to date, no single model other than the deep CNN found in [2] has been able to *simultaneously* account for retinal ganglion cell responses to both natural scenes and all four of these classes of stimuli, as well as several other classes of artificial stimuli. However, it is difficult to explain the computational mechanisms underlying the deep CNN's ability to generate these responses simply by examining the complex network in Fig. 1A. For example, why does the deep CNN fire more when a stimulus is omitted, or when a bar reverses? How can it anticipate motion to compensate for propagation delays? And why do stronger responses cause earlier firing?

These are foundational scientific questions about the retina whose answers require conceptual insights that are not afforded by the existence of a complex but highly predictive CNN alone. And even more importantly, if we could extract conceptual insights into the computational mechanisms underlying CNN responses, would these mechanisms match those used in the biological retina? Or is the deep CNN *only* accurate at the level of modelling the input-output map of the retina, while being fundamentally inaccurate at the level of underlying mechanisms? Adjudicating between these two possibilities is essential for validating whether the deep learning approach to modelling in sensory neuroscience can indeed succeed in elucidating biological neural mechanisms, which has traditionally been the gold-standard of circuit based understanding in systems neuroscience [11, 22, 23, 24].

In the following we will show how a combination of dimensionality reduction and hidden neuron or stimulus attribution can yield simplified subnetwork models of the deep CNNs response to stimuli, finding models that are consistent with prior mechanistic models with experimental support in the case of latency encoding, motion reversal, and motion anticipation. In addition, our analysis yields a new model that cures the inadequacies of previous models of the OSR. Thus our overall approach provides a new roadmap to extract mechanistic insights into deep CNN function, confirms in the case of the retina that deep CNNs do indeed learn computational mechanisms that are similar to those used in biological circuits, and yields a new experimentally testable hypothesis about retinal computation. Moreover, our results in the retina yield hope (to be tested in future combined theory and experiments) that more complex deep CNN models of higher visual cortical regions, may not only yield accurate black box models of input-output transformations, but may also yield veridical and testable hypotheses about intermediate computational mechanisms underlying these transformations, thereby potentially placing deep CNN models of sensory brain regions on firmer epistemological foundations.

## 2 From deep CNNs to neural mechanisms through model reduction

To extract understandable reduced models from the millions of parameters comprising the deep CNN in Fig. 1A and [2], we first reduce dimensionality by exploiting spatial invariance present in the artificial stimuli carefully designed to specifically probe retinal physiology (Fig.1B-E), and then carve out important sub-circuits using modern attribution methods [9, 10]. We proceed in 3 steps:

**Step (1): Quantify the importance of a model unit with integrated gradients.** The nonlinear input-output map of our deep CNN can be expressed as $r(t) = \mathcal{F}[s(t)]$, where $r(t) \in \mathbb{R}^+$ denotes the nonnegative firing rate of a ganglion cell at time bin $t$ and $s(t) \in \mathbb{R}^{50 \times 50 \times 40}$ denotes the recent spatiotemporal history of the visual stimulus spanning two dimensions of space $(x, y)$ (with 50 spatial bins in each dimension) as well as 40 preceding time bins parameterized by $\Delta t$. Thus a single component of the vector $s(t)$ is given by $s_{xy\Delta t}(t)$, which denotes the stimulus contrast at position $(x, y)$ at time $t - \Delta t$. We assume a zero contrast stimulus yields no response (i.e. $\mathcal{F}[0] = 0$). We can decompose, or attribute the response $r(t)$ to each preceding spacetime point by considering a straight path in spatiotemporal stimulus space from the zero stimulus to $s(t)$ given by $s(t; \alpha) = \alpha s(t)$ where the path parameter $\alpha$ ranges from 0 to 1 [9]. Using the line integral $\mathcal{F}[s(t; 1)] = \int_0^1 d\alpha \frac{\partial \mathcal{F}}{\partial s}\big|_{s(t,\alpha)} \cdot \frac{\partial s(t,\alpha)}{\partial \alpha}$, we obtain

$$r(t) = \mathcal{F}[s(t)] = \sum_{x=1}^{50} \sum_{y=1}^{50} \sum_{\Delta t=1}^{40} s_{xy\Delta t}(t) \int_0^1 d\alpha \frac{\partial \mathcal{F}}{\partial s_{xy\Delta t}(t)}\bigg|_{\alpha s(t)} \equiv \sum_{x=1}^{50} \sum_{y=1}^{50} \sum_{\Delta t=1}^{40} \mathcal{A}_{xy\Delta t}. \quad (1)$$

This equation represents an *exact* decomposition of the response $r(t)$ into attributions $\mathcal{A}_{xy\Delta t}$ from each preceding spacetime stimulus pixel $(x, y, \Delta t)$. Intuitively, the magnitude of $\mathcal{A}_{xy\Delta t}$ tells us how important each pixel is in generating the response, and the sign tells us whether or not turning on each pixel from 0 to $s_{xy\Delta t}(t)$ yields a net positive or negative contribution to $r(t)$. When $\mathcal{F}$ is linear, this decomposition reduces to a Taylor expansion of $\mathcal{F}$ about $s(t) = 0$. However, in the nonlinear case, this decomposition has the advantage that it is exact, while the linear Taylor expansion $r(t) \approx \frac{\partial \mathcal{F}}{\partial s(t)}\big|_{s=0} \cdot s(t)$ is only approximate. The coefficient vector $\frac{\partial \mathcal{F}}{\partial s(t)}\big|_{s=0}$ of this Taylor expansion is often thought of as the linear spacetime receptive field (RF) of the model ganglion cell, a concept that dominates sensory neuroscience. Thus choosing to employ this attribution method enables us to go beyond the dominant but imperfect notion of an RF, in order to understand *nonlinear* neural responses to arbitrary spatiotemporal stimuli. In supplementary material, we discuss how this theoretical framework of attribution to input space can be temporally extended to answer different questions about how deep networks process spatiotemporal inputs.

However, since our main focus here is model reduction, we consider instead attributing the ganglion cell response back to the first layer of hidden units, to quantify their importance. We denote by $z_{cxy}^{[1]}(t) = W_{cxy}^{[1]} \circledast s(t) + b_{cxy}$ the pre-nonlinearity activation of the layer 1 hidden units, where $W_{cxy}$ and $b_{cxy}$ are the convolutional filters and biases of a unit in channel $c$ ($c = 1, \ldots, 8$) at convolutional position $(x, y)$ (with $x, y = 1, \ldots, 36$). Now computing the line integral $\mathcal{F}[s(t; 1)] = \int_0^1 d\alpha \frac{\partial \mathcal{F}}{\partial z^{[1]}}\big|_{s(t,\alpha)} \cdot \frac{\partial z^{[1]}}{\partial \alpha}$ over the same stimulus path employed in (1) yields

$$r(t) = \sum_{x,y,c} \left[ \int_0^1 d\alpha \frac{\partial \mathcal{F}}{\partial z_{cxy}^{[1]}}\bigg|_{s(t,\alpha)} \right] (W_{cxy}^{[1]} \circledast s) = \sum_{x,y,c} [\mathcal{G}_{cxy}(s)] (W_{cxy}^{[1]} \circledast s) = \sum_{x,y,c} \mathcal{A}_{cxy}. \quad (2)$$

This represents an *exact* decomposition of the response $r(t)$ into attributions $\mathcal{A}_{cxy}$ from each subunit at the same time $t$ (since all CNN filters beyond the first layer are purely spatial). This attribution further splits into a product of $W_{cxy}^{[1]} \circledast s$, reflecting the activity of that subunit originating from spatiotemporal filtering of the preceding stimulus history, and an effective stimulus dependent weight $\mathcal{G}_{cxy}(s)$ from each subunit to the ganglion cell, reflecting how variations in subunit activity $z_{cxy}^{[1]}$ as the stimulus is turned on from 0 to $s(t)$ yield a net impact on the response $r(t)$. A positive (negative) effective weight indicates that increasing subunit activity along the stimulus path yields a net excitatory (inhibitory) effect on $r(t)$.

**Step (2): Exploiting stimulus invariances to reduce dimensionality.** The attribution of the response $r(t)$ to first layer subunits in (2) still involves $8 \times 36 \times 36 = 10,368$ attributions. We can, however, leverage the spatial uniformity of artificial stimuli used in neurophysiology experiments to

reduce this dimensionality. For example, in the OSR and latency coding, stimuli are spatially uniform, implying $W_c^{[1]} \circledast s \equiv W_{cxy}^{[1]} \circledast s$ is independent of spatial indices $(x, y)$. Thus, we can reduce the number of attributions to the number of channels via

$$r(t) = \sum_{c=1}^{8} \left( \sum_{x=1}^{36} \sum_{y=1}^{36} \mathcal{G}_{cxy}(s) \right) \cdot (W_c^{[1]} \circledast s) \equiv \sum_{c=1}^{8} \mathcal{G}_c(s) \cdot (W_c^{[1]} \circledast s) \equiv \sum_{c=1}^{8} \mathcal{A}_c. \quad (3)$$

For the moving bar in both motion reversal and motion anticipation, $W_{cx}^{[1]} \circledast s \equiv W_{cxy}^{[1]} \circledast s$ is independent of the $y$ index and we can reduce the dimensionality from 10,368 down to 288 by

$$r(t) = \sum_{c=1}^{8} \sum_{x=1}^{36} \left( \sum_{y=1}^{36} \mathcal{G}_{cxy}(s) \right) \cdot (W_{cx}^{[1]} \circledast s) \equiv \sum_{c=1}^{8} \sum_{x=1}^{36} \mathcal{G}_{cx}(s) \cdot (W_{cx}^{[1]} \circledast s) \equiv \sum_{c=1}^{8} \sum_{x=1}^{36} \mathcal{A}_{cx}. \quad (4)$$

More generally for other stimuli with no obvious spatial invariances, one could still attempt to reduce dimensionality by performing PCA or other dimensionality reduction methods on the space of hidden unit pre-activations or attributions over time. We leave this intriguing direction for future work.

**Step (3): Building reduced models from important subunits.** Finally, we can construct minimal circuit models by first identifying "important" units defined as those with large magnitude attributions $\mathcal{A}$. We then construct our reduced model as a one hidden layer neural network composed of only the important hidden units, with effective connectivity from each hidden unit to the ganglion cell determined by the effective weights $\mathcal{G}$ in (2), (3), or (4).

# 3 Results: the computational structure of retinal prediction

We now apply the systematic model reduction steps described in the previous section to each of the retinal stimuli in Fig. 1B-E. We show that in each case the reduced model yields scientific hypotheses to explain the response, often consistent with prior experimental and theoretical work, thereby validating deep CNNs as a method for veridical scientific hypothesis generation in this setting. Moreover, our approach yields integrative conceptual insights into how these diverse computations can all be *simultaneously* produced by the *same* set of hidden units.

## 3.1 Omitted stimulus response

As shown in Fig. 1B, periodic stimulus flashes trigger delayed periodic retinal responses. However, when this periodicity is violated by omitting a flash, the ganglion cell signals the violation with a large burst of firing [25, 26]. This OSR phenomenon is observed across several species including salamander [12, 13]. Interestingly, for periodic flashes in the range of 6-12Hz, the latency between the last flash before the omitted one, and the burst peak in the response, is proportional to the period of the train of flashes [12, 13], indicating the retina retains a short memory trace of this period. Moreover, pharmacological experiments suggest ON bipolar cells are required to produce the OSR [13, 17], which have been shown to correspond to the first layer hidden units in the deep CNN [1, 2].

These phenomena raise two fundamental questions: what computational mechanism causes the large amplitude burst, and how is the timing of the peak sensitive to the period of the flashes? There are two theoretical models in the literature that aim to answer these questions. One proposes that the bipolar cell activity responds to each individual flash with an oscillatory response whose period adapts to the period of the flash train [18]. However, recent direct recordings of bipolar cells suggest that such period adaptation is not present [27]. The other model claims that having dual pathways of ON and OFF bipolar cells are enough to reproduce most of the aspects of the phenomena observed in experiments [17]. However, the model only reproduces the shift of the onset of the burst, and not a shift in the peak of the burst, which has the critical predictive latency [18].

Direct model reduction (Fig. 2) of the deep CNN in Fig. 1A using the methods of section 2 yields a more sophisticated model than any prior model, comprised of *three* important pathways that combine one OFF temporal filter with two ON temporal filters. Unlike prior models, the reduced model exhibits a shift in the peak of the OSR burst as a function of the frequency of input flashes.

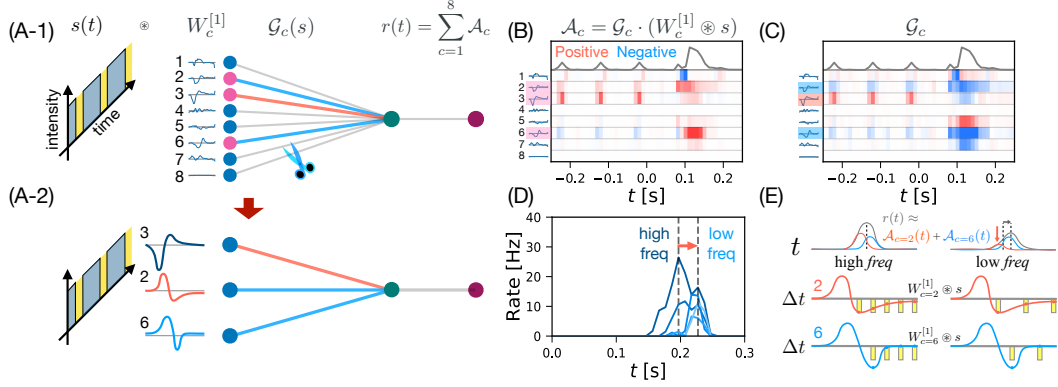

Figure 2: **Omitted stimulus response.** (A-1,2) Schematics of the model reduction procedure by only leaving three (1 OFF, 2 ON) highly contributing units. (B) Attribution for each of the cell types $\mathcal{A}_c$ over time. (C) Effective stimulus dependent weight for each of the cell types $\mathcal{G}_c$ over time. (D) The combination of the two pathways of filter 2 and 6 reproduces the period dependent latency. (E) Two ON bipolar cells are necessary to capture the predictive latency. Cell 2 with earlier peak is only active in a high-frequency regime, while the cell 6 with later peak is active independent of the frequency.

Fig. 2A presents a schematics of the model reduction steps described in (3). We first attribute a ganglion cell's response to $8$ individual channels and then average across both spatial dimensions (Fig. 2A-1) as in steps (1) and (2). Then we build a reduced model from the identified important subunits that capture essential features of the omitted stimulus response phenomenon (Fig. 2A-2). In Fig. 2B, we present the time dependence of the attribution $\mathcal{A}_c(s(t))$ in (3) for the eight channels, or cell-types. Red (blue) traces reflect positive (negative) attributions. Channel temporal filters are to the left of each attribution row and the total output response $r(t)$ is on the top row. The stimulus consists of three flashes, yielding three small responses, and then one large response after the end of the three flashes (grey line). Quantitatively, we identify that cell-type 3 dominantly explains the small responses to preceding flashes, while cell types 2 *and* 6 are necessary to explain the large burst after the flash train ends. The final set of units included in the reduced model should be the minimal set required to capture the defining features of the phenomena of interest. In the case of omitted stimulus response, the defining feature is the existence of the large amplitude burst whose peak location is sensitive to the period of the applied flashes. Once we identify the set of essential temporal filters, we then proceed to determine the sign and magnitude of contribution (excitatory or inhibitory) of the cell types. In Fig. 2C, we present the time-dependent effective weights from $\mathcal{G}_c(s(t))$ in (3) for the eight cell types, or channels. Red (blue) reflects positive (negative) weights. Given the product of the temporal filters and the weights, cell-types 2 and 6 are effectively ON cells, which cause positive ganglion cell responses to contrast increments, while cell-type 3 is an OFF cell, which is a cell type that causes positive responses to contrast decrements. Following the prescribed procedures, carving out the 3 important cell-types and effective weights yields a novel, mechanistic three pathway model of the OSR, with 1 OFF and 2 ON pathways. Unlike prior models, the reduced model exhibits a shift in the peak of the OSR burst as a function of the frequency of input flashes (with dark to light blue indicating high to low frequency variation in the flash train) as in Fig. 2D. Furthermore, the reduced model is consistent across the frequency range that produces the phenomena. Finally, model reduction yields conceptual insights into how cell-types 2 and 6 enable the timing of the burst peak to remember the period of the flash train (Fig. 2E). The top row depicts the decomposition of the overall burst response $r(t)$ (grey) into time dependent attributions $\mathcal{A}_2$ (red) and $\mathcal{A}_6$ (blue), obeying the relation $r(t) \approx \mathcal{A}_2 + \mathcal{A}_6$. Cell-type 2, which has an earlier peak in its temporal filter, preferentially causes ganglion cell responses in high-frequency flash trains (left) compared to low frequency trains (right), while cell-type 6 is equally important in both. The middle row shows the temporal filter $W_{c=2}(\Delta t)$, which has an earlier peak with a long tail, enabling it to integrate across not only the last flash, but also preceding flashes (yellow bars). Time increases into the past from left to right. Thus, the activation of this cell type 2 decreases as the flash train frequency decreases, explaining the decrease in attribution in the top row. The bottom row shows that the temporal filter $W_{c=6}(\Delta t)$ of cell type 6, in contrast, has a later peak with a rapidly decaying tail. Thus the temporal convolution $W_{c=6}(\Delta t) \circledast s(\Delta t)$ of this filter with the flash train is sensitive only to the last flash, and

is therefore independent of flash train frequency. The late peak and rapid tail explain why it supports the response at late times independent of frequency in the top row.

Thus, our systematic model reduction approach yields a new model of the OSR that cures important inadequacies of prior models. Moreover, it yields a new, experimentally testable scientific hypothesis that the OSR is an emergent property of *three* bipolar cell pathways with specific and diverse temporal filtering properties.

## 3.2 Latency coding

Rapid changes in contrast (which often occur for example right after saccades) elicit a burst of firing in retinal ganglion cells with a latency that is shorter for larger contrast changes [14] (Fig. 1C). Moreover, pharmacological studies demonstrate that both ON and OFF bipolar cells (corresponding to first layer hidden neurons in the deep CNN [1, 2]) are necessary to produce this phenomenon [19].

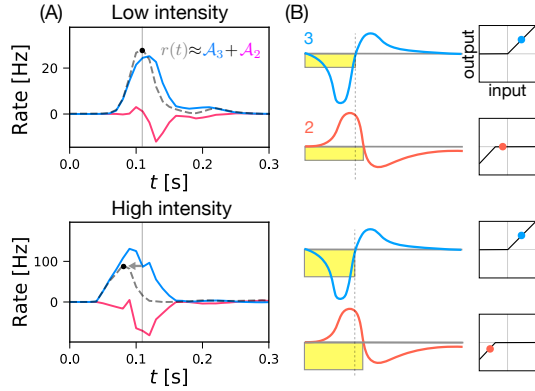

Model reduction via (3) in section 2 reveals that a single pair of slow ON and fast OFF pathways can explain the shift in the latency Fig. 3. First, under a contrast decrement, there is a strong, fast excitatory contribution from the OFF pathway. Second, as the magnitude of the contrast decrement increases, delayed inhibition from the slow ON pathway becomes stronger. This negative delayed contribution truncates excitation from the OFF pathway at late times, thereby causing a shift in the location of the total peak response to earlier times (Fig. 3). The dual pathway mechanism formed by slow ON and fast OFF bipolar cells is consistent with all existing experimental facts. Moreover, it has been previously proposed as a theory of latency coding [14, 19]. Thus this example illustrates the power of a general natural scene based deep CNN training approach, followed by model reduction, to automatically generate veridical scientific hypotheses that were previously discovered only through specialized experiments and analyses requiring significant effort [14, 19].

Figure 3: **Latency coding** (A) The decomposition of the overall response $r(t)$ (grey) into dominant attributions $\mathcal{A}_3(t)$ (blue) from an OFF pathway, and $\mathcal{A}_2(t)$ (red) from an ON pathway, obeying the relation $r(t) \approx \mathcal{A}_3 + \mathcal{A}_2$. Under a contrast decrement, the OFF pathway activated first, followed by delayed inhibitory input from the ON pathway. (B) As the amount of contrast decrement increases (yellow bars), delayed inhibition from the ON pathway (red) strengthens, which cuts off the total response in $r(t)$ at late times more strongly, thereby shifting the location of the peak of $r(t)$ to earlier times.

## 3.3 Motion reversal

As shown in Fig. 1D and [15], when a moving bar suddenly reverses its direction of motion, ganglion cells near the reversal location exhibit a sharp burst of firing. While a ganglion cell classically responds as the bar moves through its receptive field (RF) center from left to right *before* the motion reversal, the sharp burst *after* the motion reversal does *not* necessarily coincide with the spatial re-entry of the bar into the center of the RF as it moves back from right to left. Instead, the motion reversal burst response occurs at a *fixed temporal latency* relative to the time of motion reversal, for a variety of reversal locations within 110 $\mu$m of the RF center. These observations raise two fundamental questions: why does the burst even occur and why does it occur at a fixed latency?

The classical linear-nonlinear model cannot reproduce the reversal response; it only correctly reproduces the initial peak associated with the initial entry of a bar into the RF center [15]. Thus a nonlinear mechanism is required. Model reduction of the deep CNN obtained via (4) reveals that two input channels arrayed across 1D $x$ space can explain this response through a specific nonlinear mechanism (Fig. 4). Moreover, the second important channel revealed by model reduction yields a cross cell-type inhibition that explains the fixed latency (Fig. 4D). Intriguingly, this reduced model is

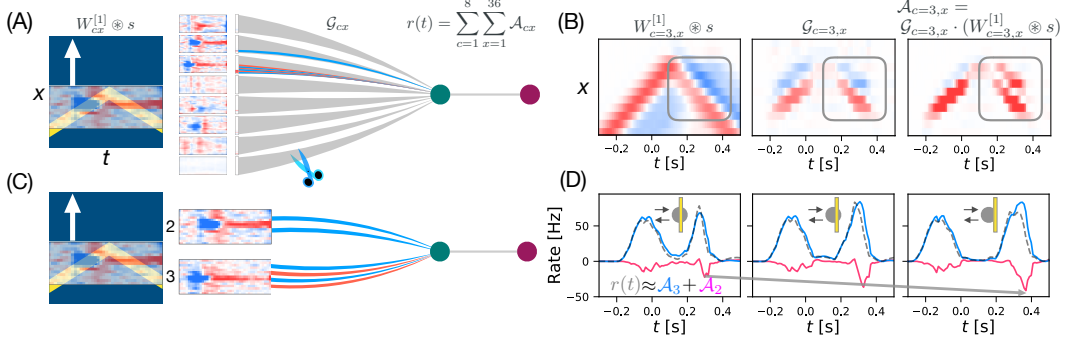

Figure 4: **Motion reversal of a moving bar.** (A) Schematics of $(x, t)$ spatiotemporal model reduction obtained via (4). By averaging over $y$ we obtain 8 cell types at 36 different $x$ positions yielding 288 units. Attribution values reveal that only cell types 2 and 3 play a dominant role in motion reversal. (B) The properties of cell-type 3 explains the existence of the burst. On the left are time-series of pre-nonlinearity activations $W^{[1]}_{c=3,x} \circledast s$ of hidden units whose RF center is at spatial position $x$. Time $t = 0$ indicates the time of motion reversal. The boxed region indicates the spatial and temporal extent of the retinal burst in response to motion reversal. The offset of the box from time $t = 0$ indicates the fixed latency. A fixed linear combination with constant coefficents of this activation cannot explain the existence of the burst due to cancellations along the vertical $x$-axis in the boxed region. However, due to downstream nonlinearities, the effective weight coefficients $\mathcal{G}_{c=3,x}$ from subunits to ganglion cell responses rapidly flip in sign (middle), and generating a burst of motion reversal response (right). (C) Schematics of the reduced model keeping only important subunits. (D) Attribution contributions from the two dominant cell types $\mathcal{A}_2$ (in pink) and $\mathcal{A}_3$ (in blue), where $\mathcal{A}_c = \sum_{x=1}^{36} \mathcal{A}_{cx}$. With only cell-type 3, the further the reversal location is from a ganglion cell's RF center, the longer we would expect it to take to generate a reversal response. However, the inhibition coming from cell type 2 increases the further away the reversal occurs, truncating the late response and thus fixing the latency of the motion reversal response.

qualitatively consistent with a recently proposed and experimentally motivated model [20] that points out the crucial role of dual pathways of ON and OFF bipolar cells.

## 3.4 Motion anticipation

As shown in Fig. 1E and [16] the retina already starts to compensate for propagation delays by advancing the retinal image of a moving bar along the direction of motion, so that the retinal image does not lag behind the instantaneous location as one might naively expect.

Model reduction of our deep CNN reveals a mechanism for this predictive tracking. First, since ganglion cell RFs have some spatial extent, a moving bar naturally triggers some ganglion cells *before* entering their RF center, yielding a leading edge of a retinal wave. What is then required for motion anticipation is some additional motion direction sensitive inhibition that cuts off the lagging edge of the wave so its peak activity shifts towards the leading edge. Indeed, model reduction reveals a computational mechanism in which one cell type feeds an excitatory signal to a ganglion cell while the other provides

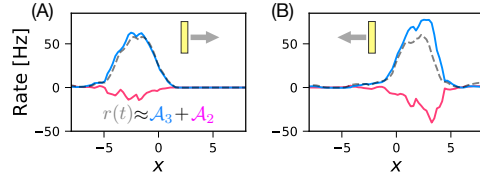

Figure 5: **Motion anticipation of a moving bar.** Contributions from the two dominant cell types. $A_2$ in pink, $A_3$ in blue, $r(t) \approx \mathcal{A}_2 + \mathcal{A}_3$ in grey, where $\mathcal{A}_c = \sum_{x=1}^{36} \mathcal{A}_{cx}$. Depending on the direction of motion of a bar, activity that lags behind the leading edge gets asymmetrically truncated by the inhibition from the cell type 2 (pink). (A) The bar is moving to the right and the inhibition (pink) is slightly stronger on the left side. (B) the bar is moving to the left and the inhibition (pink) is stronger on the right side.

direction sensitive inhibition that truncates the lagging edge. This model is qualitatively consistent with prior theoretical models that employ such direction selective inhibition to anticipate motion [16].

# 4 Discussion

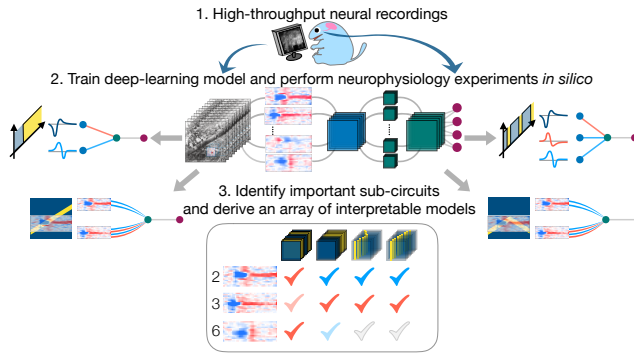

Figure 6: **A unified framework to reveal computational structure in the brain.** We outlined an automated procedure to go from large-scale neural recordings to mechanistic insights and scientific hypotheses through deep learning and model reduction. We validate our approach on the retina, demonstrating how only three cell-types with different ON/OFF and fast/slow spatiotemporal filtering properties can nonlinearly interact to *simultaneously* generate diverse retinal responses.

In summary, in the case of the retina, we have shown that complex CNN models obtained via machine learning can not only mimic sensory responses to rich natural scene stimuli, but also can serve as a powerful and automatic mechanism for generating valid scientific hypotheses about computational mechanisms in the brain, when combined with our proposed model reduction methods (Fig. 6). Applying this approach to the retina yields conceptual insights into how a *single* model consisting of multiple nonlinear pathways with diverse spatiotemporal filtering properties can explain decades of painstaking physiological studies of the retina. This suggests in some sense an inverse roadmap for experimental design in sensory neuroscience. Rather than carefully designing special artificial stimuli to probe specific sensory neural responses, and generating individual models tailored to each stimulus, one could instead fit a complex neural network model to neural responses to a rich set of ethologically relevant natural stimuli, and then apply model reduction methods to understand how different parts of a *single* model can *simultaneously* account for responses to artificial stimuli across many experiments. The interpretable mechanisms extracted from model reduction then constitute specific hypotheses that can be tested in future experiments. Moreover, the complex model itself can be used to design new stimuli, for example by searching for stimuli that yield divergent responses in the complex model, versus a simpler model of the same sensory region. Such stimulus searches could potentially elucidate functional reasons for the existence of model complexity.

In future studies, it will be interesting to conduct a systematic exploration of universality and individuality [28] in the outcome of model reduction procedures applied to deep learning models which recapitulate desired phenomena, but are obtained from different initializations, architectures, and experimental recordings. An intriguing hypothesis is that the *reduced* models required to explain specific neurobiological phenomena arise as universal computational invariants across the ensemble of deep learning models parameterized by these various design choices, while many other aspects of such deep learning models may individually vary across these choices, reflecting mere accidents of history in initialization, architecture and training.

It would also be extremely interesting to stack this model reduction procedure to obtain multilayer reduced models that extract computational mechanisms and conceptual insights into deeper CNN models of higher cortical regions. The validation of such extracted computational mechanisms would require further experimental probes of higher responses with carefully chosen stimuli, perhaps even stimuli chosen to maximize responses in the deep CNN model itself [29, 30]. Overall the success of this combined deep learning and model reduction approach to scientific inquiry in the retina, which was itself not at all *a priori* obvious before this work, sets a foundation for future studies to explore this combined approach deeper in the brain.

## Acknowledgments

We thank Daniel Fisher for insightful discussions and support. We thank the Masason foundation (HT), grants from the NEI (R01EY022933, R01EY025087, P30-EY026877) (SAB), and the Simons, James S. McDonnell foundations, and NSF Career 1845166 (SG) for funding.

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
