[Supplementary Material]

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

# Supplemental Materials

## 5   Path-integrated gradients for nonlinear dynamical systems

A specific firing burst of a ganglion cell $r(t)$ (e.g. omitted stimulus response, motion reversal etc.) is a result of spatiotemporal visual stimuli $s(t, x, y)$ processed through a non-linearly entangled web of neurons, $r(t) = \mathcal{F}[s(t, x, y)]$. Here we aim to identify and isolate the minimal neural circuit at play, in order to obtain theoretical and biophysical insights responsible for an array of retinal phenomena. Towards this goal, we need to attribute the contributions to each of the input pixels or interneurons. For a linear system, the product of the response function and an input stimulus $\chi(\Delta t, x, y)s(t - \Delta t, x, y)$ quantifies how much an input pixel changes the output. However, it is not obvious how to introduce such measure local in $(t, x, y)$ space for non-linear systems, thus raises questions: how can we, *in non-linear systems*, quantify the contributions of each of the (1) spatiotemporal input pixels, (2) internal neurons, and (3) different cell types (filters)? Here we harness attribution methods based on path-integrated gradients, first introduced in game theory [31], and then recently in machine learning [9]. We then extend the theoretical framework to a movie input with an additional dimension of time.

**Retina as a non-linear functional**
Mathematically, a mapping from a space of spatiotemporal stimulus $s(t, x, y)$ to a scalar response of a ganglion cell $r(t)$ at time $t$ can be represented by a functional,

$$r(t) = \mathcal{F}\big[s(\Delta t, x, y; t)\big]. \tag{5}$$

Here we assumed (i) causality and (ii) finite temporal memory of a ganglion cell with a cutoff at $\Delta t_m$. Thus, the input $s(\Delta t, x, y; t)$ is a subset of the entire spatiotemporal stimulus $s(t', x, y)$ in a finite region $t \in [t - \Delta t_m, t]$ and parametrized by $t$.

**Expansion of a general functional v.s. linear response theory**
The Taylor expansion of the above functional around $s(\Delta t, x, y; t) \sim 0$ is,

$$\begin{aligned}
r(t) &= \mathcal{F}\big[s(\Delta t, x, y; t)\big] \\
&= F[0] + \int d(\Delta t_1)dx_1 dy_1 \left.\frac{\delta \mathcal{F}\big[s(t)\big]}{\delta s(\Delta t_1, x_1, y_1; t)}\right|_{s=0} s(\Delta t_1, x_1, y_1; t) + \mathcal{O}(s^2)
\end{aligned} \tag{6}$$

In comparison, the linear response theory takes the form of,

$$r(t) = \mathcal{F}\big[s(\Delta t, x, y; t)\big] = \mathcal{F}[0] + \int d(\Delta t)dx dy \chi(\Delta t, x, y)s(\Delta t, x, y; t), \tag{7}$$

where $\chi$ is the linear response function (e.g. receptive field around zero stimuli).
Two main approximations are made in the linear response theory; (i) no stimulus $s$ dependence of $\frac{\delta \mathcal{F}}{\delta s}$, (ii) truncation of the Taylor expansion at the first order. These two assumptions are valid only when $|s(\Delta t, x, y; t)| \ll 1$ holds everywhere. However, this condition hardly holds in visual systems in action, and the above conventional formulation is due to theoretical and experimental limitations rather than a well-justified approximation as in Figure 7A for schematics.

Figure 7: **How important is an input pixel or a cell type (filter)?** (A) [Linear theory: red] Conventional linear theory, such as classical receptive field, extrapolates linear trend to all over the input stimulus space. Deep learning model trained on natural scenes is able to compute local gradient, "instantaneous receptive field", given an input of structured stimuli with high efficiency. However, the instantaneous receptive field only explains the local property of the non-linear function that does not necessalily reflect the global structure. [*IntegratedGradients* in 1D: blue] By integrating the local gradients captured by a deep learning model over input axis, we can obtain a more acculate description of the non-linear function. (B) The path-integrated gradients becomes dependents on the path in higher dimensions ($D > 1$), but straight path is shown to satisfy desired axioms [9]. Schematic illustrates the case of two-dimensions.

While both of the biological retina and the deep learning model gives $r(t) = \mathcal{F}\big[s(\Delta t, x, y; t)\big]$ upon a presentation of a stimulus, the stimulus-dependent instantaneous receptive field

$$\left.\frac{\delta \mathcal{F}}{\delta s}\right|_s \tag{8}$$

is easily computable in the deep learning model. Below, we harness the quantity $\delta\mathcal{F}/\delta s|_s$ to investigate where a ganglion cell is looking at during a burst of neural firings.

**Review: Path integrated gradients for an image input**

Here we first review a case of an image input $r = \mathcal{F}[s(x, y)]$, where $s(x, y)$ represents a two dimensional image. For a general path $\gamma(\alpha)$ parameterized by $\alpha$,

$$r(\alpha = 1) - r(\alpha = 0) = \mathcal{F}[s(x, y, \alpha = 1)] - \mathcal{F}[s(x, y, \alpha = 0)]$$
$$= \int dx dy \int_0^1 d\alpha \left[\frac{\delta \mathcal{F}(x, y; \alpha)}{\delta s(x, y; \alpha)} \frac{\partial s(x, y; \alpha)}{\partial \alpha}\right] \equiv \int dx dy A_{\gamma(\alpha)}(x, y). \tag{9}$$

Sundararajan *et al.* [9] proved that a straight path $s(x, y; \alpha) = \alpha s(x, y)$ satisfies an array of desirable axioms: Sensitivity, Insensitivity, Linearity, Preservation, Implementation invariance, Completeness, and Symmetry. In that case, the above further simplifies to,

$$\mathcal{A}_{\text{IG}}(x, y) = s(x, y) \int d\alpha \frac{\mathcal{F}(x, y; \alpha)}{\delta s(x, y; \alpha)}. \tag{10}$$

See Figure 7 for schematics.

**Space-time attribution by path integrated gradients for a movie input**

Here we try to extend the theoretical framework of the above *IntegratedGradients* to the case of a movie input. In addition to the spatial components $(x, y)$, a movie input has an additional dimension of time $t$.

In this case, the attribution score $\mathcal{A}(\Delta t, x, y; t)$ can be computed by taking a path integral parametrized by $\alpha$,

$$r(t) = \mathcal{F}\big[s(\Delta t, x, y; t)\big] - \mathcal{F}\big[s(\Delta t, x, y; \alpha = 0)\big]$$
$$= \int d(\Delta t) dx dy \left[\int d\alpha \left.\frac{\delta \mathcal{F}\big[s(\Delta t, x, y; \alpha)\big]}{\delta s(\Delta t, x, y; \alpha)}\right|_{s(\alpha)} \frac{s(\Delta t, x, y; \alpha)}{\partial \alpha}\right] \tag{11}$$
$$\equiv \int d(\Delta t) dx dy \mathcal{A}(\Delta t, x, y; t).$$

Thus, attribution score can be defined as

$$\mathcal{A}(\Delta t, xx, y; t) \equiv \left[\int d\alpha \left.\frac{\delta \mathcal{F}\big[s(\Delta t, x, y; \alpha)\big]}{\delta s(\Delta t, x, y; \alpha)}\right|_{s(\alpha)} \frac{s(\Delta t, x, y; \alpha)}{\partial \alpha}\right]. \tag{12}$$

For a linear system, where $\frac{\delta \mathcal{F}}{\delta s}$ is independent of $s$, we can simply replace it by a linear response function $\frac{\delta \mathcal{F}}{\delta s}|_{s=0} = \chi$, and the attribution score reduces to a product of the linear response function and the difference between the current stimulus and the baseline,

$$\mathcal{A}(\Delta t, x, y; t) = \chi(\Delta t, x, y; t)\big(s(\Delta t, x, y; t) - s(\Delta t, x, y; \alpha = 0)\big). \tag{13}$$

However, for a non-linear system, $\frac{\delta \mathcal{F}}{\delta s}|_s$ depends on $s$ and the attribution score $\mathcal{A}(\Delta t, x, y)$ depends on the space of function that we integrate over. Namely, each "path" corresponds to a different attribution method and provides different explanations to different questions. In particular, for a movie input, the history of stimuli forms a path naturally parametrized by time "$t$".

**IntegratedGradients: Effective linear response function for non-linear systems**

Here we focus on the straight path,

$$s(\Delta t, x, y; \alpha) = \alpha s(\Delta t, x, y; t). \tag{14}$$

In this case, the integral of the gradients act as an effective linear response function for a nonlinear system as below,

$$\mathcal{A}_{IG}(\Delta t, x, y; t) \equiv \left[\int d\alpha \frac{\delta \mathcal{F}\big[\alpha s(\Delta t, x, y; t)\big]}{\delta \big(\alpha s(\Delta t, x, y; t)\big)}\right] s(\Delta t, x, y; t) \tag{15}$$
$$= \chi(\Delta t, x, y; t)s(\Delta t, x, y; t) = \chi_{[s]}(t, t', x, y)s(t', x, y).$$

By computing the attribution $\mathcal{A}(\Delta t, x, y; t)$ over the full temporal region, we can obtain the full attribution $\mathcal{A}(t, t', x, y)$. Note that the function now depends both on $t$ and $t'$ due to nonlinearity. Furthermore, we can reformulate the above equation into two forms to answer specific questions raised below.

Figure 8: **Straight path integration from the null stimulus** We computed *IntegratedGradients* for a periodic flashes with omitted stimulus by taking a straight path integration of gradients from the null stimulus $s_0 = 0$ to an actual stimulus $s_1$. (A) (Left) Attribution map on response time $t$ v.s input time $t'$ space. Causality limits the attribution in the region of $t' < t$. (Right) Attribution on input stimuli $\int_{t'}^{\infty} dt \mathcal{A}(t, t')$ quantifies the total amount of output created by each of the pixels of an input movie. (B) (Left) Attribution map on response time $t$ v.s. integration time $\Delta t$ space. (Right) Attribution on integration time $\Delta t$ integrated over a single periodic burst triggered by flashes (blue) and omitted stimulus response (red). The peak of attribution for the flash response (blue) is placed earlier than the peak of attribution for the omitted stimulus response (orange). This is consistent with our finding that the responses for periodic flashes are caused by a fast OFF bipolar cell, while the omitted stimulus response is triggered by slow ON bipolar cells. (See main text.)

Figure 9: **Straight path integration between two stimuli** We computed *IntegratedGradients* to discriminate between two stimuli continuing periodic flashes $s_0$, and periodic flashes with omitted stimulus $s_1$. (A) (Left) Attribution map on response time $t$ v.s input time $t'$ space. Causality limits the attribution in the region of $t' < t$. (Right) Attribution on input stimuli $\int_{t'}^{\infty} dt \mathcal{A}(t, t')$ quantifies the total amount of output created by each of the pixels of ano input movie. Strikingly, the attribution concentrates at the location of the first omitted stimulus among the four omitted stimulus. (B) (Left) Attribution map on response time $t$ v.s. integration time $\Delta t$ space. (Right) Attribution on integration time $\Delta t$ integrated over a single periodic burst triggered by flashes (blue) and omitted stimulus response (orange).

**Q1. Which pixel in the spatiotemporal input stimulus "$(t, x, y)$" is responsible for a ganglion cell's response?**

$$\int_{t_0}^{t_1} dt r(t) = \int dt' dx dy \left( \int_{t_0}^{t_1} dt \mathcal{A}(t, t', x, y) \right) \tag{16}$$

See Figure 8 for an example with OSR.

**Q2. Which part of the spatiotemporal kernel is "$(\Delta t, x, y)$" responsible for a ganglion cell's response?**

$$\int_{t_0}^{t_1} dt r(t) = \int dx dy \int_{t_0}^{t_1} dt \int_{t_0 - \Delta t_m}^{t_1} dt' \mathcal{A}(t, t', x, y) \delta\big((t - t') - \Delta t\big) = \int d\Delta t dx dy \left( \int_{t_0}^{t_1} dt \mathcal{A}(t, t - \Delta t, x, y) \right) \tag{17}$$

See Figure 9 for an example with OSR.

# 6 Extension to deeper CNNs without spatial invariances in stimuli

In the main text, we demonstrated how we can reduce CNNs with three layers to multi-pathways linear non-linear models, specifically to validate deep CNNs in the retina where we had neurobiological ground truth. Here, we discuss one possible way to extend our method to deeper CNNs of depth $D$ processing natural movies through a dynamic programming (DP) approach that works backwards from layer $D$ to layer 1. First, note a natural movie of limited duration without spatial invariances is still well approximated by a low dimensional trajectory in both pixel space and every hidden layer. Let $K$ be the max dimensionality for spatial input patterns for any channel

in any layer. Then the basic idea is to attribute the response in layer $D$ to the $K$ dimensional space of inputs to each channel in layer $D-1$ using integrated gradients. We first find the important channels in layer $D-1$ using methods presented in the main text. Then we recursively iterate via the same method to layer $D-2$ and so on down to the pixel layer. Because of the DP-like nature of our algorithm, the computational complexity (after dimensionality reduction to $K$) is $O(DKC)$ where $C$ is the max number of channels in a layer, and not exponential in $D$. The end result is a set of important channels in each layer, along with, for each important channel $\leq K$ linear combinations of neurons that matter for generating the response in layer $D$.