[Reviews · NeurIPS 2019]

Reviewer 1



I really liked this paper. However, as someone not yet familiar with the attribution method, I found the algorithmic details quite sparse. I would like a description of how the attribution *algorithm* works, and not just the mathematics underpinning it. For example, from Eq. 2, it looks to me like attribution could be done by using regression between r(t) and the W*s convolution, to find G_{cxy}. Is that what the authors did? Was there any regularization to that regression? How were the regularization parameters chosen? These details matter for readers who might want to use this approach in their own work. It would also be nice for the authors to share their code; I'd like to mess around with these methods myself, and I suspect the same will be true of other readers.

Reviewer 2



Originality: The paper applies integrates gradient methods to DNNs used for predicting neural activity in the retina for identifying important subunits to build reduced models, which lend them self to interpretation. This is a original contribution with potential in the field. The paper does feel like a follow up to ref. 2, and Figure 1 seems almost a complete reproduction of a figure in ref. 2. Quality: The analysis presented in the paper is of uniformly high quality, but the paper is so strangely structured, that it dampens my enthusiasm significantly, despite the potential of the method. The Introduction is extremely long, at the expense of a very superficial discussion of the individual parts of the results, which contain the really interesting new bits of information. Also, many interesting derivations, explanations and discussions are in the supplement, for example the justification why the methods works or which information it provides. Also, e.g. the figure caption of Fig. 2 is almost a whole page, containing information which rather should be discussed in the main text. As it is, the papers focusses almost exclusively on the discussion of the “new models” but fails to highlight its methodological contributions. In their reply, the authors promised to restructure the paper significantly; I am not certain, however, whether this can still be fit within the scope of the review process at NeurIPS, or will require so significant revisions, that it will have to be submitted elsewhere for rereview. Clarity: Together with the supplement, the methods are clear and most model explanations make sense. Significance: Potentially large, but paper not suited for NeurIPS in its present form.

Reviewer 3



This manuscript aims to attack an interesting problem, namely how could one obtain mechanistic insights from the CNN model fit to the neural responses. The writing is generally clear, although it would benefit to tone down some of the statements to more accurately reflect the real contributions. Overall, the manuscript could be an interesting contribution to the field. However, I am skeptical about various claims made in the paper. The main issues I have with this manuscript are three-fold: 1.the results is rather incremental relatively to ref [2] and [9,10]. 2.it is unclear to me to what extent the insights as claimed in Section 3 are novel, and to what extent they could not be obtained by taking a much simpler approach. 3.it is unclear to what extent the proposed algorithmic approach could apply to other systems, such as high level visual cortex. Major * Questions/comments about the proposed algorithms: the first step is based on the attribution methods developed in [ref 9,10], and a simple extension to incorporate the temporal domain. This step is largely incremental. the second step reduces dimensionality by exploring stimulus invariance. How could this generalize to more complex stimuli where the stimulus invariance properties are unclear? step 3 involves constructing reduced model from “important” subunits. The number of selected subunit is not discussed and seems to be arbitrary. Practically, how to set this number for a given problem? For example, unit 1 in Fig2B,C also seems to be important, but it is not selected. *Results on OSR For OSR results, the study identified three channels, A2,A3,A6. How do these channels map onto the physiologically defined cell types in retina? Is there experimental evidence in any form that would support this model prediction? The asymmetry in the physiologically observed nonlinearity in ON/OFF pathways [Chichilnisky& Kalmar, 2002; Kareem et al., 2003] are not accounted in the model. Could it be possible that by taking these asymmetry into account, one just need two channels, rather than three to account for OSR? Ref: Chichilnisky, E. J., and Rachel S. Kalmar. "Functional asymmetries in ON and OFF ganglion cells of primate retina." Journal of Neuroscience 22.7 (2002): 2737-2747. Zaghloul, Kareem A., Kwabena Boahen, and Jonathan B. Demb. "Different circuits for ON and OFF retinal ganglion cells cause different contrast sensitivities." Journal of Neuroscience 23.7 (2003): 2645-2654. *Interpretation of the results Regarding the results shown in Section 3.2-3.4, to what extent doe they provide novel insights? Could it be that given the response characteristics of the ON/OFF channels and the observed phenomena, these is effectively only one mechanism that could account for the observed data? In that case, it seems that one would not need to go through the exercise of reducing from a deep network model to obtain such results; one could directly constrain the input and output then find the transformation that could link the two. Relatedly, for the results presented regarding the four applications, how much of these one could already obtain by using the standard RF +nonlinearity analysis? *Applicability/complexity of the approach is it possible to apply the proposed approach to analyze deeper neural networks? The manuscript analyzed a 3-layer CNN. Intuitively, the complexity of the analysis might scale exponentially with increasing number of layers. It is unclear if the proposed dimension reduction approach would still be effective in these cases, e.g., CNN for the object recognition. Note that for object recognition task, the simple invariance assumption as assumed in the paper to achieve dimension reduction might also be violated. Is there any claim could be made about the complexity of the analysis for deeper networks? * The special role of A2&A3 By examining all the results together, it seems that most of the effects are readily accounted by cell types A2 and A3. I feel that this should be more carefully discussed somewhere in the paper. Also is it possible to use a CNN model with 2 or 3 channels rather than 8 to account for the results? Other comments - I found the title to be a bit mis-leading,, in particular “prediction”. The manuscript does not directly the connection of the phenomena to the function of prediction. The same concern applies to the abstract (line 14-16). - line 34, how is the “computational mechanism” defined? - line 36, what does “non-trivial response” mean? - “burst” is used in various places. However, the CNN model doesn’t contain any spiking mechanism. What does “burst” mean in the context of the CNN model? - Step 3 involves reconstructing a network with one hidden layer neural network. Would it be expressive enough to explain a variety of computational mechanisms? For example, consider the task of object recognition. - line 62-64. it should be made clear that all the results in fig.1 have been previously shown in ref [2]. - line85-87, These are simply rephrase the phenomena. Maybe cut these? - line 134, “far beyond the imperfect but dominant”, maybe could tone this down a bit? In principle, nothing is perfect, including the RF analysis. %%%% After discussions and seeing the authors' rebuttal letter, I still have concerns about the applicability of the proposed method to other problems. However, I think the authors' response did help clean up a bunch of issues, so I am increasing my score from 5 to 6.

Reviewer 4



In this paper, the authors present an approach to extract mechanistic insights from deep CNNs trained to recreate retinal responses to natural scenes. Specifically, the authors use a combination of model attribution methods and dimensionality reduction to uncover cell types from the CNN that explain nonlinear retinal responses to four classes of stimuli. The authors uncover mechanistic understanding of latency coding, motion reversal responses, and motion anticipation in the retina that fits with prior scientific findings and models. The authors uncover a new model for omitted stimulus responses that is better able to explain retinal responses than prior models and forms a testable hypothesis. There are important limitations to the work presented here. The methods depend on artificial stimuli with spatial invariances and it is unclear that these methods will extend to more complex stimuli. The authors state that perhaps other stimuli could be reduced using PCA or similar methods but this paper would be more impactful if the authors demonstrated this or even discussed possible future directions in more detail. Additionally, the authors mostly recreate known retina phenomena and mechanisms. They do yield a new testable model of OSR but since this is not tested yet, it is unknown if their approach yielded new knowledge about the retina. Providing some experimental follow-up on the scientific hypothesis generated by this work would be extremely impactful. I think the paper should acknowledge and address these limitations/caveats more thoroughly - the work felt overstated at times. Despite this, I think this paper is novel and significant. Moving the relatively new field of fitting neurons with deep networks beyond simply improving predictions to gaining scientific understanding is extremely important and this paper is a solid start to these efforts. It is encouraging the the deep CNN was trained on natural scenes and not specifically on the four classes of stimuli. The paper is well-written and relatively easy to understand. Minor comments: I disagree slightly with the emphasis that deep networks fit to neural responses must yield computational mechanisms matching intermediate computations in the brain to be useful. This is one particularly good avenue of research but deep networks predictive of neural responses could be used to find testable experimental phenomena (like those presented in this paper) through artificial experiments or to better understand neural computations at a more abstract level. The paper is unclear whether the authors find the same 3 cell types to explain the responses for each stimulus - this is mentioned in the Figure 6 caption but is not emphasized elsewhere. All figures should be larger for clarity. Figure 1 B-E are not very helpful without more explanation for readers unfamiliar with these concepts. The colors in Figure 2E are hard to distinguish - maybe use different colors or a wider range of blue shades? EDIT: I've read the author response - it was thorough but did not convince me to change my score.

[Author Response · NeurIPS 2019]

We thank all reviewers for their careful reviews and many positive comments, including **R1**: "really valuable to the neuro community", "gives a roadmap for using NNs...to tell us how brains work", "I really liked this paper"; **R4**: "this paper is novel and significant," "well-written and relatively easy to understand"; **R3**: "could be an interesting contribution to the field." Even the most negative reviewer **R2** stated: "a highly original contribution with huge potential in the field," "the analysis...is of uniformly high quality, "with the supplement, the methods are clear and most model explanations make sense." We now address major reviewer concerns and clarify our contributions, as detailed below:

**Extension to deeper CNNs without spatial invariances in stimuli (R2,R3,R4)**: While we demonstrated a novel application of attribution methods to model reduction of 1-hidden layer CNNs, specifically to validate deep CNNs in the retina where we had neurobiological ground truth, we can easily extend our method to deeper CNNs of depth $D$ processing natural movies through a dynamic programming (DP) approach that works backwards from layer $D$ to layer 1. First, note a natural movie of limited duration without spatial invariances is still well approximated by a low dimensional trajectory in both pixel space and every hidden layer. Let $K$ be the max dimensionality for spatial input patterns for any channel in any layer. Then the basic idea is to attribute the response in layer $D$ to the $K$ dimensional space of inputs to each channel in layer $D-1$ using integrated gradients. We first find the important channels in layer $D-1$ using methods in our paper. Then we recursively iterate via the same method to layer $D-2$ and so on down to the pixel layer. Because of the DP-like nature of our algorithm, the computational complexity (after dimensionality reduction to $K$) is $O(DKC)$ where $C$ is the max number of channels in a layer, and not exponential in $D$ as **R3** worried. The end result is a set of important channels in each layer, along with, for each important channel $\leq K$ linear combinations of neurons that matter for generating the response in layer $D$. We are actively pursuing this method in deeper networks, but we will share pseudocode for this algorithm in a revised version before acceptance to NeurIPS. However, consistent with **R2,R3,R4** we feel completing this program is well beyond the scope of this paper, especially since neurobiological ground truth is missing for higher areas. But we hope our success in the retina and the extendability of our approach to deeper networks, will provide a great roadmap for neuroscience as recognized by **R1**.

**Experimental evidence for our new model of omitted stimulus response (OSR) (R3)**: As shown quantitatively in [2], the model subunits match bipolar cells (BCs), and the 3 in the OSR correspond to fast OFF, fast ON and slow ON BCs, thus mapping directly to biological pathways. Furthermore, multiple BC types can connect to a ganglion cell (GC) (Asari and Meister 2012). Thus our new model is basically consistent with known anatomy. However, we leave *further* physiological validation of our model, beyond successfully generating the OSR, to future work, which would require painstaking experiments to perturb BC pathways and observe GC responses. We believe it is already a substantial contribution to show our approach *automatically* extracts validated models for 3 stimuli, and provides a new, experimentally testable model for a fourth (we will add suggested experiments to the paper). The main aim of our paper is to publish our new hypotheses in order to stimulate multiple retina labs worldwide to tackle the difficult neurophysiology experiments. In this manner, our theory could generate new experimental progress in future work.

**Simpler approaches do not suffice (R3)**: A single linear receptive field (RF) plus a nonlinearity (LN model) cannot account for any of the 4 stimuli (indeed that is precisely why these stimuli are interesting). References from **R3** show that ON/OFF pathways differ in their threshold as well as timing, and optimized two-pathway LN models could partially capture the OSR [17] but *cannot* produce sufficient frequency-dependent shifting of the latency [18]. Thus the reported asymmetries cannot produce the observed OSR response, and our new finding is that three pathway LN models can.

**Clarifying our contribution beyond previous work (R3)**: While building on a deep retina network from the authors of [2], that work did not provide conceptual understanding of *how* the network generated responses to 4 highly structured stimuli, and *whether* it generated those responses the same way the retina did. We provided such an explanation, showing only 3 of 8 channels were required to generate responses to all 4 stimuli in an interpretable manner, thereby demonstrating a single approach (natural scenes -> deep CNN -> model reduction) that can *simultaneously* discover what was previously only discovered piecemeal across $\geq 10$ papers. We feel this yields a major advance in providing a "roadmap for neuroscience" (**R1**). Moreover, our method is primarily a *novel application* of attribution methods in [9,10] to model reduction in neuroscience with validation in a biological circuit. From the NeurIPS call for papers, such *application* papers are squarely within conference scope, and major advances in attribution methodology should not be required for acceptance since that direction is orthogonal to our application to model reduction in neuroscience. We will however revise to tone-down, discuss limitations, and clarify specific contributions (**R3,4**).

**Revising the text (R2)** We will follow **R2**'s excellent suggestions; we will shorten the intro, expand results, move info from Fig. 2 caption to text, and provide more background on integrated gradients in the main, using the extra page for the camera-ready. We note **R2** gave the lowest score (4 compared to 8 (**R1**) and 7 (**R4**)), despite being very positive (**R2**: "highly original contribution with huge potential," "with the supplement, the methods are clear"). We hope, given our restructuring, **R2** will be convinced that the revised version will be acceptable.

**Other comments (R1-R4)** Though we cannot address all remaining less major comments in the author response due to lack of space, we assure reviewers we can easily do so in the revision. We are grateful for your excellent suggestions.

[Meta-Review · NeurIPS 2019]

Dear authors, congrats on the acceptance-- this paper was discussed extensively and controversially. The reviewers provided multiple comments and a lot of feedback, and it will be of critical importance that you revise the manuscript accordingly (and as you promised in your rebuttal)- in particular, this would require significant changes in the exposition, and also a more clear description of the novelty of the study in light of previous work.